# Nutritional Status Measurement Instruments for Diabetes: A Systematic Psychometric Review

**DOI:** 10.3390/ijerph17165719

**Published:** 2020-08-07

**Authors:** Pedro Montagut-Martínez, David Pérez-Cruzado, José Joaquín García-Arenas

**Affiliations:** 1Health Sciences PhD Program, Campus de los Jerónimos, Universidad Católica de Murcia UCAM, 30107 Murcia, Spain; pmontagut2@alu.ucam.edu; 2Department of Occupational Therapy, Campus de los Jerónimos, Universidad Catolica de Murcia UCAM, 30107 Murcia, Spain; jjgarcia2@ucam.edu; 3Institute of Biomedicine of Malaga (IBIMA), 29071 Málaga, Spain

**Keywords:** nutritional status, diabetes mellitus, questionnaire

## Abstract

Background: Diabetes is a serious chronic disease associated with a large number of complications and an increased risk of premature death. A dietary evaluation is of utmost importance for health promotion, disease prevention and individual treatment plans in patients with diabetes. Methods: An exhaustive search was carried out in various databases—Medline, Web of Science, Open Gray Cochrane Library and Consensus-based Standards for the selection of health status Measurement Instruments (COSMIN)—for systematic review of the measurement properties of instruments that evaluate the dietary intake of people with diabetes mellitus type 1 and/or 2 according to COSMIN standards. Results: Seven instruments were identified. There was no instrument measuring nutritional status for which all the psychometric properties were evaluated. The methodological quality for each of the psychometric properties evaluated was ‘inadequate’ or ‘doubtful’ for all instruments. The Food Frequency Questionnaire (FFQ) evaluated the most psychometric characteristics and with a better score in terms of quality of the evidence. Conclusions: Several instruments have been developed for the evaluation of dietary intake in people with diabetes. Evaluation of this construct is very useful, both in clinical practice and in research, requiring new knowledge in this area. The FFQ is the best instrument available to assess dietary intake in people with diabetes.

## 1. Introduction

Diabetes is a serious chronic disease associated with a large number of complications and an increased risk of premature death [1]. Due to these complications, almost half of the deaths (46.1%) were in adults under 60 years old, the working age group [2]. Diabetes is a challenging public health problem, with serious health consequences and healthcare costs associated with an unhealthy lifestyle (vision loss, amputation, renal failure and cardiovascular diseases) [3,4]. Diabetes management requires pharmacological interventions and lifestyle changes, these being associated with a decreased risk of death [5,6,7]. Maintaining an adequate diet positively influences the evolution of diabetes and the appearance of complications [8,9]. Patients with diabetes usually face difficulties in identifying an adequate diet, both in quality and quantity [10]. Food selection and dietary pattern are influenced by the patient’s knowledge of a recommended diet [11]; therefore, dietary evaluation is of utmost importance for health promotion, disease prevention and individual treatment plans in patients with diabetes [10,12]. For this reason, there is a need for briefing tools that allow health professionals with a general knowledge of nutrition to assess the dietary intake of their patients and establish dietary goals based on the evaluation obtained [13,14]. In this sense, there is a wide variety of validated instruments that can be used for a fast and accurate evaluation [15,16]. These tools do not match the precision of a detailed evaluation by a professional, but they can be used to promote change in diet when chronic conditions are managed [17]. Currently, there are no systematic reviews in the literature of this great variety [18] of instruments in which an evaluation of the psychometric properties and methodological quality of the instruments based on COSMIN (Consensus-based Standards for the selection of health status Measurement Instruments) has been carried out [19,20]. Taking this fact into account, a systematic review was carried out to identify the instruments for evaluating nutritional status in patients with diabetes mellitus (DM) type 1 and/or 2 (DM1/DM2), assess their psychometric properties and assess the methodological quality of the validation studies of these instruments. The objectives of this systematic review were to identify instruments to measure the nutritional status of patients with DM1 and/or DM2 and their psychometric properties, and evaluate the methodological quality of the validation studies of these instruments.

## 2. Materials and Methods

### 2.1. Design

A systematic review of the measurement properties of instruments that evaluate the dietary intake of people with DM1 and/or DM2 was carried out according to COSMIN methodology for systematic reviews of Patient-Reported Outcome Measures (PROMs) (published on February 2018) standards [19,20].

### 2.2. Databases and Search Strategy

An exhaustive search in various databases—Medline (through Pubmed), Web of Science, Open Gray, Cochrane Library and COSMIN—was carried out during February 2020 by two independent reviewers.

In order to locate the maximum possible number of tools and their different versions, no dates were set. To carry out this search, the Medical Subject Headings (MeSH) terms were combined with synonyms and similar terms (Table 1), adapting said strategy to each of the search systems of the different databases. In addition, the reference lists of the included articles were checked to identify other relevant documents. The search strategy was carried out following COSMIN methodology for systematic reviews of Patient-Reported Outcome Measures (PROMs).

### 2.3. Inclusion/Exclusion Criteria

The inclusion criteria were: articles published without a time limit that included an instrument that evaluated nutritional status in patients with DM1 and/or DM2, evaluation of at least one measurement property of those reflected in the COSMIN checklist (content validity, structural validity, internal consistency, reliability, measurement error, hypotheses testing for construct validity, cross-cultural validity/measurement invariance, criterion validity and responsiveness) and articles written in English or Spanish. Editorials, biographies, speeches, comments and conferences, case reports, practice guides, doctoral theses, opinion articles or publications in which selective methodology was not used were excluded.

Titles and abstracts were independently selected by two reviewers. The full article was retrieved when decisions on the abstract and title alone could not be made. Discrepancies were resolved by discussion with a third reviewer.

### 2.4. Assessment of Methodological Quality

The included studies were evaluated by Pedro Montagut-Martínez (P.M.-M.), José Joaquín García-Arenas (J.J.G.-A.) and David Pérez-Cruzado (D.P.-C.) using the COSMIN four-point checklist for systematic reviews [20].

The COSMIN checklist presents nine boxes with 5–18 elements, each box providing a methodological quality score for content validity (property that assesses whether each of the included items is relevant and complete), structural validity (measures the degree to which the scores of an instrument adequately show the dimensionality of the construct to be measured), internal consistency (degree of correlation between the different elements), cross-cultural validity (investigation of a phenomenon or theory that is tested in at least two different cultures), measurement error (difference between a subject’s empirical score and its true score), reliability (measurement stability), criterion validity (degree to which an instrument’s scores correspond to a ‘gold standard’), hypothesis testing (property is analysed to check if hypotheses were formulated a priori) and responsiveness (ability of a measure to show changes over time).

The evaluated content in each box can be described as ‘very good’, ‘adequate’, ‘doubtful’ and ‘inadequate’. The score for each psychometric property was obtained by taking the lowest score of any element for each property [20].

### 2.5. Data Extraction

Three authors independently assessed the methodological quality of the studies, with discrepancies resolved by consensus. All articles were selected by review of title, abstract and full text. Information was extracted from each article regarding the following aspects: name of the tool used, study population, setting, description of the tool, evaluated psychometric properties, COSMIN score and statistical values of the evaluated psychometric properties.

### 2.6. Synthesis of Data

The data on the psychometric properties of the articles that met the inclusion criteria were scored according to the COSMIN checklist [20]. The results of each study for each of the properties of measurement were rated according to the Prinsen criteria [21], with each result rated as ‘positive’ (+), ‘negative’ (−) or ‘indeterminate’ (?). Finally, the quality of the evidence and strength of the recommendations were rated ‘high’, ‘moderate’, ‘low’ or ‘very low’ using the Grading of Recommendations Assessment, Development and Evaluation (GRADE) system [22]. Scores for each of the psychometric properties in each article were attributed by three independent reviewers, and differences were resolved by discussion and consensus.

## 3. Results

A total of 3538 articles were identified, of which 2217 titles and abstracts were examined. Of these, 2195 articles were excluded after evaluation. Subsequently, 22 full-text articles were evaluated, and 8 articles were excluded after reviewing the full text [11,23,24,25,26,27,28,29]. Thus, a total of 14 studies were included in this systematic review. The flow chart is shown in Figure 1.

Seven instruments were identified that were specifically designed to assess dietary intake in patients with DM1 and/or DM2: the Diabetes Knowledge and Behaviour (DKB) questionnaire [30], the Food Frequency Questionnaire (FFQ) [31,32,33,34,35,36,37,38,39,40], the Perceived Dietary Adherence Questionnaire (PDAQ) [41], the UK Diabetes and Diet Questionnaire (UKDDQ) [42], the self-developed Dietary Knowledge Questionnaire (DKQ) [40], the Diabetes Mellitus Knowledge questionnaire (DMK) [40] and the Motiv. Diaf-DM2 questionnaire (MDDM2) [43]. In most cases, the original language used was English, although we can find several of these questionnaires adapted to other versions: Brazilian [32], Mali [33], Austrian [34], Korean [35] and Chinese [31] versions of the FFQ, an Arabic version of the FFQ, DKQ and DMK [40] and a Spanish version of the MDDM2 [43]. The FFQ varied in the number of items and was compared with a ‘gold standard’ (Weighed Dietary Record) in the majority of studies where it was used. Most of these studies were carried out in hospital, community and primary care settings, in patients with diabetes mellitus type 1 and/or 2 (Table 2).

### 3.1. Psychometric Properties

The main properties evaluated were content validity, criterion validity, hypothesis testing and responsiveness. The results of the four-point COSMIN checklist of instruments for measuring nutritional status in patients with DM1 and/or DM2 are shown in Table 3.

Table 4 shows data on the instruments for measuring dietary intake using the COSMIN checklist and GRADE approach.

#### 3.1.1. Diabetes Knowledge and Behaviour Questionnaire (DKB)

The psychometric properties of this questionnaire were evaluated in a single study [30]. The score for internal consistency was ‘inadequate’, with Cronbach’s alpha between 0.59 and 0.90; the hypothesis test was rated as ‘doubtful’. This 12-item questionnaire on food frequency showed moderate correlation values for the dietary evaluation of total calories (*r* = 0.48–0.64) and calories due to fat (*r* = 0.41–0.65) but a negative correlation between a simple question related to the frequency of fruit consumption and the fat index in Europeans (*r* = 0.25) and Maori (*r* = −0.33).

#### 3.1.2. Food Frequency Questionnaire (FFQ)

This questionnaire was evaluated in ten studies [31,32,33,34,35,36,37,38,39,40] but content validity was assessed in only two of these [33,40], scoring between ‘inadequate’ and ‘doubtful’, mainly due to the lack of validity assessment of content by professionals and patients with diabetes. Structural validity was evaluated in the study by Sami et al. [40], obtaining a ‘doubtful’ score, and exploratory factor analysis (EFA) resulted in a five-factor solution. Cronbach’s alpha was between 0.782 and 0.908 [40]. Test–retest reliability was measured for two studies, with the score being ‘inadequate’, with ICC values of 0.98 [36] and 0.24–0.71 [37]. In most of the studies, the FFQ was compared with a gold standard (Weighed Dietary Record), and the score reflected in criterion validity was found to be ‘inadequate’ for all these studies [31,32,34,35,37,38,39]; the correlations varied between *r* = 0.23 [34] and *r* = 0.8 [37]. The hypothesis test was rated as between ‘inadequate’ and ‘doubtful’. For the study by Coulibaly et al. [33], there was a significant correlation between protein intake estimated using the FFQ and the 48-h withdrawals (*r* = 0.63); for Liese et al. [36], the mean correlations, adjusted for measurement error, of the food and nutrient groups between the FFQ and true habitual intake were r = 0.41 and *r* = 0.38, respectively; finally, for Sami et al. [40] the correlations between the different items of the FFQ ranged between *r* = 0.632 and *r* = 0.928.

#### 3.1.3. Perceived Dietary Adherence Questionnaire (PDAQ)

The PDAQ was only evaluated in one study [41], in which content validity was rated as ‘doubtful’. Cronbach’s alpha was 0.78 and ICC = 0.78. For the hypothesis test, the correlation coefficients for PDAQ items versus 24-h withdrawals ranged from *r* = 0.46 to *r* = 0.11.

#### 3.1.4. UK Diabetes and Diet Questionnaire (UKDDQ)

This questionnaire was only evaluated in one study [42], where content validity was rated as ‘inadequate’ due to the lack of validity assessment of content by professionals and patients with diabetes. Regarding its reliability, ICC = 0.90 was obtained. For the hypothesis test, there were moderate correlations between the total UKDDQ scores and the food diaries with regard to alcohol consumption (ANOVA = 0.71) and ‘breakfast cereals’ (ANOVA = 0.70).

#### 3.1.5. Self-Developed Dietary Knowledge Questionnaire (DKQ)

A single study assessed the psychometric properties of this questionnaire [40], in which content validity was rated as ‘doubtful’, as was structural validity, where EFA resulted in a five-factor solution. The five factors were labelled ‘food selection’, ‘health impact’, ‘healthy choices’, ‘food restriction’ and ‘food categorization’. In this study, the internal consistency produced a Cronbach’s alpha of 0.869. For the hypothesis test, the total correlation for the DKQ between its different items ranged from *r* = 0.364 to *r* = 0.588.

#### 3.1.6. Diabetes Mellitus Knowledge Questionnaire (DMK)

The DMK was evaluated in only one of the studies [40], the content validity being ‘doubtful’, as well as the structural validity, where EFA resulted in a five-factor solution. These five factors were labelled ‘food selection’, ‘health impact’, ‘healthy choices’, ‘food restriction’ and ‘food categorization’. For this study, Cronbach’s alpha was 0.891. In the hypothesis test, the total correlation for the DMK between its different items ranged from *r* = 0.358 to *r* = 0.529.

#### 3.1.7. Motiv.Diaf-DM2 Questionnaire (MDDM2)

This questionnaire was included in only one of the studies [43], in which content validity and structural validity were rated as ‘inadequate’. The MDDM2 showed a two-dimensional structure (dietary intake and adherence to physical activity), with a Cronbach’s alpha of 0.756 for the first factor and 0.821 for the second factor. For the hypothesis test, the first dimension did not obtain significant correlations with the evaluated variables (basic psychological needs, resilience and glycated haemoglobin). Scores for the second dimension (adherence to physical activity) are in line with basic psychological needs (*r* = 0.281) and resistance (*r* = 0.216); in addition, a relatively moderate relationship was found with glycated haemoglobin (*r* = −0.182).

## 4. Discussion

This is the first systematic review in which instruments to assess nutritional status in patients with DM1 or DM2 are identified and their psychometric properties and methodological quality are evaluated based on COSMIN [19,20]. In this study, seven instruments specifically designed to assess nutritional status in diabetic patients were identified from 14 studies.

Different factors, such as scope, the population where the instrument will be used, its dimensions, the number of items and the evidence shown in the evaluation of each psychometric property must be considered by clinicians and researchers to decide which instrument is the best to measure nutritional status. In addition, the language and culture of the original version of the instrument must be taken into account. Systematic reviews about nutritional status have been published focusing on the elderly and on children, but these instruments have been not specifically validated for people DM1 or DM2; the use of these instruments in this population could give us different information about their nutritional status [44,45]. The food frequency questionnaire included in both systematic reviews [44,45] has been validated in people with DM1 or DM2 and their psychometric properties have been shown in the present study [31,32,33,34,35,36,37,38,39,40].

There was no instrument measuring nutritional status for which all the psychometric properties were evaluated. Cross-cultural validity and measurement error were the only two properties that were not present in any of the seven instruments selected for review, reflecting the limitations of nutritional status. The psychometric properties evaluated in most of the studies were content validity and criterion validity, hypothesis testing and responsiveness; criterion validity was completely evaluated for the FFQ because in seven of the ten studies, it was compared with a ‘gold standard’ [31,32,34,35,37,38,39].

The methodological quality of the studies for each evaluated psychometric property was not as expected, being ‘inadequate’ or ‘doubtful’ for all the instruments due to the lack of content validity for evaluating the relevance of the items in the study population. With regard to internal consistency, factor analysis was not performed, internal consistency was not calculated for each dimension separately and there was a lack of description of the percentages and management of missing values. For hypothesis testing, the consequences of the hypotheses were not clear or were not formulated a priori even though it was possible to deduce what was expected. There was a lack of description of the management of missing values. Finally, for structural validity, reliability and criterion validity there was a lack of description of the percentages and management of the missing values. The differences in values of psychometric properties for the included instruments could be explained by the fact that in each study, the setting and the characteristics of the population were different. Despite these differences, the different questionnaires included different questions about dietary intake that could influence the reliability and validity values.

The DKB [30], PDAQ [41], UKDDQ [42], DKQ [40], DMK [40] and MDDM2 [43] were only evaluated in one study. Additional studies are required to evaluate the psychometric properties of these instruments.

The quality of the evidence was ‘low’ or ‘very low’ for all the measurement properties for the instruments included in this systematic review. Content validity, structural validity, and internal consistency are considered to be the most important measurement properties [19,21]. Content validity was present in six of the seven instruments (FFQ [33,40], PDAQ [41], UKDDQ [42], DKQ, DMK [40] and MDDM2 [43]). The quality of the evidence was ‘low’ or ‘very low’ for all of these instruments; for the UKDDQ [42], quality was rated as ‘±’, indicating that the items included are inconsistent or indeterminate with respect to the construct being measured. With regard to structural validity, again the quality of the evidence was ‘low’ or ‘very low’ for four of the instruments: FFQ, DKQ, DMK [40] and MDDM2 [43]. A five-factor structure was obtained for the FFQ, DKQ and DMK [40], but there were only two resulting factors for the MDDM2 [43].

Finally, a ‘low’ or ‘very low’ rating was obtained for all the instruments that evaluated internal consistency. In most of the tools used to assess nutritional status in patients with diabetes (FFQ [40], PDAQ [41], DKQ, DMK [40] and MDDM2 [43]), the quality of the results was rated as ‘+’ (Cronbach’s alpha = 0.70), whereas for the DKB [30] it was ‘-’ (Cronbach’s alpha = 0.70).

The FFQ [31,32,33,34,35,36,37,38,39,40] has been included in the largest volume of studies (10 studies in total) in which more psychometric characteristics have been evaluated and has demonstrated better scores in terms of the quality of the evidence. This questionnaire has been adapted to people with DM1 or DM2, including specific food that should be measured in this population. This makes this questionnaire the most widely validated and recommended for use in evaluating the dietary intake of people with diabetes.

### Strengths and Limitations

This is the most current systematic review to identify, evaluate and summarize the evidence regarding instruments measuring nutritional status in patients with DM1 or DM2. The review was performed in accordance with COSMIN standards, which ensured that an appropriate method was used, following the recommendations of the experts [19,21]. Although a meticulous search was carried out in various databases, it is possible that not all the instruments available for measurement of nutritional status have been identified in this review due to the search strategy limitation to only articles published in English and Spanish. Articles on instruments developed and used in other languages may not have been identified.

## 5. Conclusions

Various instruments have been developed for the evaluation of dietary intake in people with diabetes. Evaluation of this construct is very useful both in clinical practice and in research, requiring new knowledge in this area. The evaluation of dietary intake is essential because it allows the identification and evaluation of interventions focused on improving nutrition in people with diabetes. The FFQ is the best instrument available to assess dietary intake in people with diabetes with different versions in number of items. A greater number of studies with high methodological quality is necessary, in which all the psychometric properties of the existing instruments are analysed and the instruments are adapted to other cultures.

## Figures and Tables

**Figure 1 ijerph-17-05719-f001:**
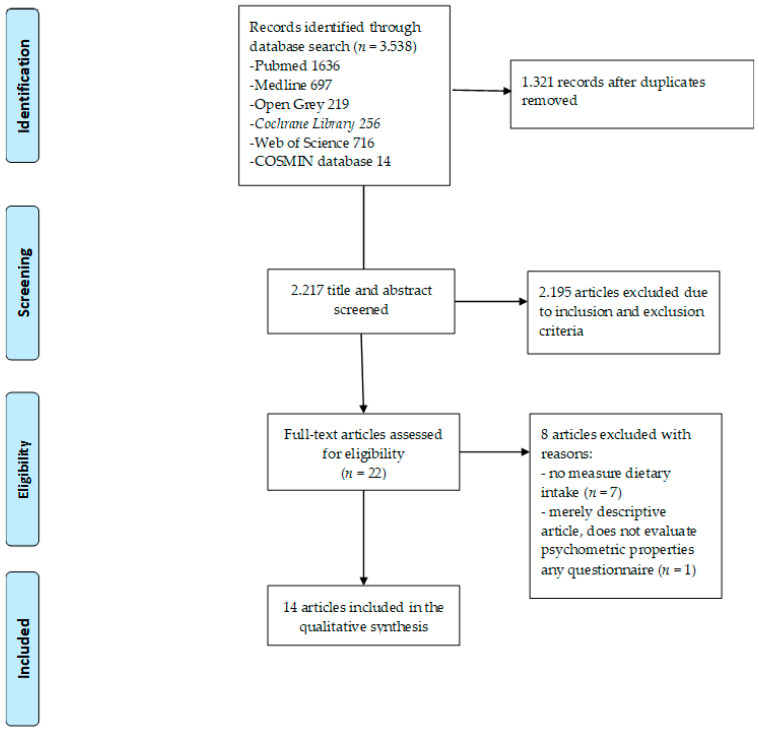
Flow chart diagram of the study selection.

**Table 1 ijerph-17-05719-t001:** Search strategy.

1 (“Diabetes mellitus” [MeSH] OR “diabetes” [All Fields])
2 (“nutritional status”[MeSH] OR “nutrition”[All Fields] OR “nutrition status”[All Fields] OR “eating” [MeSH] OR “food intake” [All Fields] OR “food frequency” [All Fields])
3 (“Instrument”[tiab] OR “instruments”[tiab] OR “measure” [tiab] OR “measures” [tiab] OR “questionnaire”[tiab] OR “questionnaires”[tiab] OR “scale”[tiab] OR “scales”[tiab] OR “tool”[tiab] OR “tools”[tiab] OR “survey” [tiab] OR “test” [tiab])
4 (“Instrumentation”[sh] OR “methods”[sh] OR “Validation Studies”[pt] OR “Comparative Study”[pt] OR “psychometrics”[MeSH] OR psychometr*[tiab] OR clinimetr*[tw] OR clinometr*[tw] OR “outcome assessment (health care)”[MeSH] OR “outcome assessment”[tiab] OR “outcome measure*”[tw] OR “observer variation”[MeSH] OR “observer variation”[tiab] OR “Health Status Indicators”[Mesh] OR “reproducibility of results”[MeSH] OR “reproducib*”[tiab] OR “discriminant analysis”[MeSH] OR “reliab*”[tiab] OR “unreliab*”[tiab] OR “valid*”[tiab] OR “coefficient”[tiab] OR “homogeneity”[tiab] OR “homogeneous”[tiab] OR ““internal consistency””[tiab] OR (“cronbach*”[tiab] AND (“alpha”[tiab] OR “alphas”[tiab])) OR (item[tiab] AND (correlation*[tiab] OR selection*[tiab] OR reduction*[tiab])) OR “agreement”[tiab] OR “precision”[tiab] OR “imprecision”[tiab] OR “precise values”[tiab] OR “test-retest”[tiab] OR(“test”[tiab] AND “retest”[tiab]) OR(“reliab*” [tiab] AND (“test”[tiab] OR “retest”[tiab])) OR“stability”[tiab] OR “interrater”[tiab] OR “inter-rater”[tiab] OR “intrarater”[tiab] OR “intra-rater”[tiab] OR “intertester”[tiab] OR “inter-tester”[tiab] OR “intratester”[tiab] OR “intra-tester”[tiab] OR “interobserver”[tiab] OR “inter-observer”[tiab] OR “intraobserver”[tiab] OR “intraobserver”[tiab] OR “intertechnician”[tiab] OR “inter-technician”[tiab] OR “intratechnician”[tiab] OR “intra-technician”[tiab] OR “interexaminer”[tiab] OR “inter-examiner”[tiab] OR “intraexaminer”[tiab] OR “intra-examiner”[tiab] OR “interassay”[tiab] OR “inter-assay”[tiab] OR “intraassay”[tiab] OR “intra-assay”[tiab] OR “interindividual”[tiab] OR “inter-individual”[tiab] OR “intraindividual”[tiab] OR “intra-individual”[tiab] OR “interparticipant”[tiab] OR “inter-participant”[tiab] OR “intraparticipant”[tiab] OR “intra-participant”[tiab] OR “kappa”[tiab] OR “kappa’s”[tiab] OR “kappas”[tiab] OR “repeatab*”[tiab] OR ((“replicab*”[tiab] OR “repeated”[tiab]) AND (“measure”[tiab] OR “measures”[tiab] OR “findings”[tiab] OR “result”[tiab] OR “results”[tiab] OR “test”[tiab] OR “tests”[tiab])) OR “generaliza*”[tiab] OR “generalisa*”[tiab] OR “concordance”[tiab] OR (“intraclass”[tiab] AND “correlation*”[tiab]) OR “discriminative”[tiab] OR “known group”[tiab] OR “factor analysis”[tiab] OR “factor analyses”[tiab] OR “dimension*”[tiab] OR “subscale*”[tiab] OR (“multitrait”[tiab] AND “scaling”[tiab] AND (“analysis”[tiab] OR “analyses”[tiab])) OR “item discriminant”[tiab] OR “interscale correlation*”[tiab] OR error[tiab] OR errors[tiab] OR “individual variability”[tiab] OR (“variability”[tiab] AND (“analysis”[tiab] OR “values”[tiab])) OR (“uncertainty”[tiab] AND (“measurement”[tiab] OR “measuring”[tiab])) OR “standard error of measurement”[tiab] OR “sensitiv*”[tiab] OR “responsive*”[tiab] OR ((“minimal”[tiab] OR “minimally”[tiab] OR “clinical”[tiab] OR “clinically”[tiab]) AND (“important”[tiab] OR “significant”[tiab] OR “detectable”[tiab]) AND (“change”[tiab] OR “difference”[tiab])) OR (“small*”[tiab] AND (“real”[tiab] OR “detectable”[tiab]) AND (“change”[tiab] OR “difference”[tiab])) OR “meaningful change” [tiab] OR “ceiling effect”[tiab] OR “floor effect”[tiab] OR “Item response model”[tiab] OR “IRT”[tiab] OR “Rasch”[tiab] OR “Differential item functioning”[tiab] OR DIF[tiab] OR “computer adaptive testing”[tiab] OR “item bank”[tiab] OR “cross-cultural equivalence”[tiab])
5 #1 AND #2 AND #3 AND #4
6 “Protocol”[Publication Type] OR “addresses”[Publication Type] OR “biography”[Publication Type] OR “case reports”[Publication Type] OR “comment”[Publication Type] OR “editorial”[Publication Type] OR “congresses” [Publication Type] OR “consensus development conference”[Publication Type] OR “consensus development conference”[Publication Type] OR “practice guideline”[Publication Type]) OR “suffering from”[tiab] OR “animals”[MeSH]
7 #5 NOT #6
8 FILTER: Language (English and Spanish)
9 FILTER: Species (Humans)

**Table 2 ijerph-17-05719-t002:** Studies characteristics and description of status measurement instrument in DM.

Study (Author and Year)	Population/Type of Mellitus Diabetes	Setting	Instrument Description	Measurement Properties	COSMIN Score	Measurement Values
Diabetes Knowledge and Behaviour Questionnaire (DKB)						
Simmons et al., (1994)	397 adults with type 2 diabetes	New Zealand: Patients were recruited from the community	DKB includes five open-ended questions, a four-point and five-point Likert scale, and 47 closed questions. The questions are grouped into 10 stems with 3–6 true/false answers. For its definitive application, the incorrect answer is assigned a negative score of −1, the correct answer is a score of +1 and “does not know” is scored with 0.	1-Internal consistency 2-Hypotheses testing 3-Responsiveness	1-Inadequate 2-Doubtful 3-Doubtful	α: 0.59–0.90 #total calories (*r* = 0.48–0.64) # calories due to fat (*r* = 0.41–0.65) Negative correlations with the frequency of fruit consumption (*r* = (−0.25)–(−0.33))
Food Frequency Questionnaire (FFQ)						
Riley et al., (1995)	84 patients with type 1 diabetes	Australia: Patients were randomly selected from a population-based insulin-treated diabetes register.	The questionnaire typically includes questions on 80 to 120 food and beverage items. In this article, the final version of the FFQ consists of 153 food items.	1-Criterion validity 2-Responsiveness	1-Inadequate 2-Inadequate	# 2-day weighed dietary (*r* = 0.38–0.60) # true usual dietary intake (*r* = 0.60)
Coulibaly et al., (2008)	57 patients with type 2 diabetes	Mali: Primary health-care services.	In this article, the final version of the FFQ consists of 53 food items.	1-Content validity 2-Hypotheses testing 3-Responsiveness	1-Inadequate 2- Inadequate 3-Doubtful	# 48 h recall (*r* = 0.63)
Hong et al., (2010)	85 patients with type 2 diabetes	Korea: Patients were recruited from Korean National Diabetes Program (KNDP)	In this article, the final version of the FFQ consists of 85 food items.	1-Criterion validity 2-Responsiveness	1-Inadequate 2-Inadequate	# energy (*r* = 0.74) # iron (*r* = 0.27) The Kappa values for energy, carbohydrate, protein, fat and calcium were 0.54, 0.37, 0.36, 0.46, and 0.19, respectively
Aguiar et al., (2013)	88 patients with type 2 diabetes	South of Brazil: Hospital de Clínicas de Porto Alegre. Out-patients	In this article, the final version of the FFQ consists of 98 food items.	1-Criterion validity 2-Responsiveness	1-Inadequate 2-Doubtful	# WDR for most nutrients
Luevano-Contreras et al., (2013)	30 patients with type 2 diabetes	USA: University of Illinois	In this article, the final version of the FFQ consists of 90 food items.	1-Reliability 2-Criterion validity 3-Responsiveness	1-Inadequate 2-Inadequate 3-Inadequate	ICC = 0.98 # FR time 1 (*r* = 0.68) # FR time 2 (*r* = 0.80)
Farukuoye et al., (2014)	27 nondiabetic relatives of patients with DM2, 66 patients with diabetes (32 patients with DM2 and 34 with DM1 diabetes) and 30 nondiabetic healthy individuals	Vienna: Diabetes Outpatient Service of the 1st Medical Department of Hanusch Hospital, Teaching Hospital of Medical University of Vienna, and from a local Physiotherapy Service.	In this article, the final version of the FFQ consists of 107 food items.	1-Criterion validity2-Responsiveness	1-Inadequate 2-Inadequate	# 7DR (*r* = 0.23–0.72)
Liese et al., (2014)	172 patients with type 1 diabetes	USA: University of North Carolina Nutrition Obesity Research Center.	In this article, the final version of the FFQ consists of 85 food items.	1-Reliability 2-Hypotheses testing 3-Responsiveness	1-Inadequate 2-Doubtful 3-Doubtful	ICC both FFQ,(*r* = 0.24–0.71) # between the items of FFQ (*r* = 0.38–0.41)
Petersen et al., (2015)	67 patients with type 1 and 2 diabetes	Australia. Patients were recruited from the community	In this article, the final version of the FFQ consists of 74 food items.	1-Criterion validity 2-Responsiveness	1-Inadequate 2-Inadequate	# WFR # Food intake
Sami et al., (2017)	132 patients with type 2 diabetes	Saudi Arabia	In this article, the final version of the FFQ consists of 99 food items.	1-Content Validity 2-Structural Validity 3-Internal Consistency 4-Hypotheses testing 5-Responsiveness	1-Doubtful 2-Doubtful 3-Doubtful 4-Inadequate 5-Inadequate	EFA resulted in five-factor solution with eigenvalues greater than 1. α = 0.782–0.908 # 24-HDRs (*r* = 0.58–0.66)
Meng-Chuan et al., (2018)	126 patients with type 2 diabetes	Taiwan: Kaohsiung Medical University Hospital	In this article, the final version of the FFQ consists of 45 food items.	1-Criterion validity 2-Responsiveness	1-Inadequate 2-Inadequate	# protein (*r* = 0.65) #fat (*r* = 0.58) #carbohydrate (*r* = 0.64) #fiber (*r* = 0.66)
Perceived Dietary Adherence Questionnaire (PDAQ)						
Asaad et al., (2015)	73 patients with type 2 diabetes	Canada: Patients were recruited from the community	Nine-item questionnaires with scores ranging from lowest 0 to highest 7 based on a seven-point Likert scale. Higher scores reflect higher adherence except for items 4 and 9, which reflect unhealthy choices (foods high in sugar or fat).	1-Content validity 2-Internal consistency 3-Reliability 4-Hypotheses testing 5-Responsiveness	1-Doubtful 2-Inadequate 3-Inadequate 4-Doubtful 5-Doubtful	α = 0.78 ICC = 0.78 # 24-HDR (*r* = 0.11–0.46)
UK Diabetes and Diet Questionnaire (UKDDQ)						
England et al., (2016)	177 patients with type 2 diabetes	United Kingdom (Southwest England): Patients were recruited into STAMP-2: Sedentary time and metabolic health in people with (or at risk of) type 2 diabetes	The UK Diabetes and Diet Questionnaire (UKDDQ) consists of 25 items; of those 25 items, 20 contribute to the overall score. Each of the 20 items has six categories for the participant to choose from, which corresponds to the frequency of consumption of the participant for that particular item, the score for each item ranges from 0–5 (with 0 being healthier and 5 less healthy).	1-Content validity 2-Reliability 3-Hypotheses testing4-Responsiveness	1-Inadequate 2-Doubtful 3-Doubtful 4-Inadequate	ICC = 0.90 Total scores from the UKDDQ and food diaries compared well ICC = 0.54.
The self-developed Dietary Knowledge Questionnaire (DKQ/DK)						
Sami et al., (2017)	132 patients with type 2 diabetes	Saudi Arabia	The self-prepared dietary knowledge questionnaire (DKQ) used in this research consists of 20 multiple-choice questions (MCQ). The answers are coded as 1 = correct and 0 = incorrect, and I don’t know. The score ranges from 0 to 20; a higher DK level is indicated by a higher score.	1-Content Validity 2-Structural Validity 3-Internal Consistency 4-Hypotheses testing 5-Responsiveness	1-Doubtful 2-Doubtful 3-Doubtful 4-Inadequate 5- Inadequate	EFA resulted in five-factor solution with eigenvalues greater than 1 α = 0.869 # between items (*r* = 0.364–0.588)
Diabetes Mellitus Knowledge Questionnaire (DMK)						
Sami et al., (2017)	132 patients with type 2 diabetes	Saudi Arabia	The new version of DMK questionnaire used in this research comprised 30 questions. Responses are coded as 1 = yes and 0 = no, and I don’t know.	1-Content Validity 2-Structural Validity 3-Internal Consistency 4-Hypotheses testing 5-Responsiveness	1-Doubtful 2-Doubtful 3-Doubtful 4-Inadequate 5-Inadequate	EFA resulted in five-factor solution with eigenvalues greater than 1 α = 0.891 # between items (*r* = 0.358–0.529)
Motiv.Diaf-DM2 Questionnaire (MDDM2)						
Martín Payo et al., (2018)	206 patients with type 2 diabetes	Spain: Primary care services	Motivate. Diaf-DM2 is made up of three blocks including sociodemographic variables, type of motivation of the patients performing physical activity and resilience. This questionnaire consists of four items in Likert format on a scale of 1 (It does not describe me at all) to 5 (It describes me very well).	1-Content validity 2-Structural validity 3-Internal consistency 4-Reliability 5-Hypotheses testing 6-Responsiveness	1-Inadequate 2-Inadequate 3-Inadequate 4-Inadequate 5-Doubtful 6-Doubtful	EFA resulted in two-dimensional instrument. A = 0.756–0.821 # between factors (*r* = 0.604–0.638)

*r*: Pearson correlations; #: Significant correlations. Abbreviations: (DM) Diabetes Mellitus; (DKB) Diabetes Knowledge and Behaviour Questionnaire; (FFQ) Food Frequency Questionnaire; (KNDP) Korean National Diabetes Program; (WDR) Weighed Diet Record; (USA) United States of America; (ICC) Intraclass Correlation Coefficient; (7DR) 7-day food record; (WFR) Weighed Food Record; (EFA) Exploratory Factor Analysis; (24-HDR) 24-h dietary recalls; (PDAQ) Perceived Dietary Adherence Questionnaire; (UKDDQ) United Kingdom Diabetes and Diet Questionnaire; (STAMP-2) Sedentary Time and Metabolic Health in People with type 2 Diabetes study; (DKQ) The self-developed Dietary Knowledge Questionnaire; (MCQ) multiple-choice questions; (DK) Dietary Knowledge; (DMK) Diabetes Mellitus Knowledge Questionnaire; (MDDM2) Motiv.Diaf-DM2 Questionnaire.

**Table 3 ijerph-17-05719-t003:** Methodological quality of the studies and quality of results reported per measurement property, instrument and study.

Instrument	Article	Content Validity		Structural Validity		Internal Consistency		Reliability		Criterion Validity		Hypotheses Testing		Responsiveness	
		M	Q	M	Q	M	Q	M	Q	M	Q	M	Q	M	Q
DKB	Simmons et al., (1994)					I	−					D	?	D	?
FFQ	Riley et al., (1995)									I	−			I	+
	Coulibaly et al., (2008)	I	?									I	?	D	?
	Hong et al., (2010)									I	−			I	+
	Aguiar et al., (2013)									I	−			D	+
	Luevano-Contreras et al., (2013)							I	+	I	−			I	+
	Farukuoye et al., (2014)									I	−			I	+
	Liese et al., (2014)							I	−			D	?	D	?
	Petersen et al., (2015)									I	+			I	+
	Sami et al., (2017)	D	−	D	?	D	+					I	+	I	+
	Meng-Chuan et al., (2018)									I	−			I	+
PDAQ	Asaad et al., (2015)	D	−			I	+	I	+			D	?	D	?
UKDDQ	England et al., (2016)	I	+ −					D	+			D	?	I	?
DKQ/(DK)	Sami et al., (2017)	D	−	D	?	D	+					I	+	I	+
DMK	Sami et al., (2017)	D	−	D	?	D	+					I	+	I	+
MDDM2	Martín Payo et al., (2018)	I	?	I	−	I	+	I	−			D	?	D	?

The cross-cultural validity/measurement invariance and measurement error were omitted in the table because they were not evaluated in any studies. M: Methodological quality of the study rated as I = Inadequate, D = Doubtful; empty boxes = not reported. Q: Quality of the results rated as + = positive rating, ? = indeterminate rating, ± = inconsistent rating, − = negative rating; empty boxes = not reported.

**Table 4 ijerph-17-05719-t004:** Summary of findings per measurement property and instrument.

Instrument	Content Validity			Structural Validity			Internal Consistency			Reliability			Criterion Validity			Hypotheses Testing			Responsiveness		
	M	Q	QE	M	Q	QE	M	Q	QE	M	Q	QE	M	Q	QE	M	Q	QE	M	Q	QE
DKB							I	−	VL							D	?	L	D	?	L
FFQ	I	−	VL	D	?	VL	D	+	L	I	−	VL	I	−	L	I	?	L	I	?	L
PDAQ	D	−	L				I	+	VL	I	+	VL				D	?	L	D	?	L
UKDDQ	I	+ −	VL							D	+	L				D	?	L	I	?	VL
DKQ/DK	D	−	L	D	?	L	D	+	L							I	+	VL	I	+	VL
DMK	D	−	L	D	?	L	D	+	L							I	+	VL	I	+	VL
MDDM2	I	?	VL	I	−	VL	I	+	VL	I	−	VL				D	?	L	D	?	L

The cross-cultural validity/measurement invariance and measurement error were omitted in the table because they were not evaluated in any studies. M: Methodological quality of the study rated as I = Inadequate, D = Doubtful; empty boxes = not reported. Q: Quality of the results rated as + = positive rating, ? = indeterminate rating, ± = inconsistent rating, − = negative rating; empty boxes = not reported. QE: Quality of evidence rated as, L = Low, VL = Very low.

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
