# Peer review of "Nutritional Status Measurement Instruments for Diabetes: A Systematic Psychometric Review"

_ijerph, 2020, doi:10.3390/ijerph17165719_

Round 1

Reviewer 1 Report

Interesting systematic summary of nutritional status measurement instruments for 2 diabetes. The topic is of medical, societal, and societal value because of the pandemic character of diabetes.

The conclusions are relevant and confirm the usefulness of the evaluation of dietary intake in people with diabetes for both doctors-practitioners dealing with diabetic patients and researchers in the field, but also the lack of the existence of the one recommended for general use.

The manuscript is not easy to follow because of the specific character of the methodology used to review the existing literature.

1.Table 1. Search strategy- is completely unreadable (without commas, semicolons, appearing and disappearing quotation marks the information is misleading; OR/AND can change the meaning)

2. Table 2 requires structuring and shortening (important info in a compact form, avoid full sentences) 

3. Descriptions 3.1.1-3.1.7 too closely repeat the content of Table 2

4. The collected data on the instruments for measuring dietary intake using the COSMIN checklist and GRADE approach (Table 4) suggest that most of these instruments do not fulfill the strict criteria of analysis, including the FFQ.

Author Response

ITEMIZED LIST OF THE REVIEWERS COMMENTS

Manuscript ID: ijerph-847417

We would like to thank the Reviewers for their thoughtful and constructive comments. We have considered all suggestions and have incorporated them into the revised manuscript. Changes to the original manuscript are with “track changes”. We believe our manuscript is stronger as a result of the modifications. An itemized point-by-point response to the Reviewers’ comments is presented below. 

REVIEWER'S REPORT

Interesting systematic summary of nutritional status measurement instruments for 2 diabetes. The topic is of medical, societal, and societal value because of the pandemic character of diabetes.The conclusions are relevant and confirm the usefulness of the evaluation of dietary intake in people with diabetes for both doctors-practitioners dealing with diabetic patients and researchers in the field, but also the lack of the existence of the one recommended for general use. The manuscript is not easy to follow because of the specific character of the methodology used to review the existing literature.

1.Table 1. Search strategy- is completely unreadable (without commas, semicolons, appearing and disappearing quotation marks the information is misleading; OR/AND can change the meaning)

Authors’s Answer: Thank you very much for the time and effort spent reviewing this document. We have improved the search strategy showed in Table 1 for a better comprehension for future readers.

  1. Table 2 requires structuring and shortening (important info in a compact form, avoid full sentences)

Authors’s Answer: Thank you very much for your suggestion. We have shortened the Table 2, with a homogeneous format and avoiding full sentences.

  1. Descriptions 3.1.1-3.1.7 too closely repeat the content of Table 2

Authors’s Answer: Thank you very much for your suggestion. According to your suggestion and after the changes made in Table 2, this section include a different information with more details about psychometric properties of the instruments.

  1. The collected data on the instruments for measuring dietary intake using the COSMIN checklist and GRADE approach (Table 4) suggest that most of these instruments do not fulfill the strict criteria of analysis, including the FFQ.

Authors’s Answer: Thank you very much for your comment. According to your comment, the results of the present study shows that the psychometric properties of the included instruments do not have a good methodological quality. Although any exclusion criteria about the methodological quality of the instruments.

Reviewer 2 Report

IJERPH_847417
Review
Nutritional status measurement instruments for diabetes: A systematic psychometric review
Reviewer 1 comment´s:
This manuscript talks about the Diabetes disease and as dietary evaluation is of utmost 13 importance for health promotion, disease prevention and individual treatment plans in patients 14 with diabetes. Method: exhaustive search was carried out in various databases that evaluate the dietary intake of people with diabetes mellitus type 1 and/or 2 according to COSMIN and PRISMA standards. Conclusions: The FFQ is the best instrument available to 25 assess dietary intake in people with diabetes.
The comments are described below:
Reviewer 1 comment´s:
-Line 77: Table 1. Search strategy.
I recommend that authors place the information as if it were a table following the “Instructions for Authors”, or as a figure, but this does not fully understand the information collected when searching. You can consult other manuscripts published in the same journal that serves as a reference.
-Line 170-182: Where is the [38] reference??
-Line 204: Please, organize all tables in a homogeneous format as table 3 or 4.
Table 2. Studies characteristics and description of status measurement instrument in DM.
-Line 215: Table 3. Methodological quality of the studies and quality of results reported per measurement property, instrument and study.
-Line 220: Table 4. Summary of findings per measurement property and instrument.
-Table 3 and 4: Aguiar et al. (2013), there isn´t in the References Section
2
-Please verify that all the authors' citations are in the references section with their corresponding numbering, and in the tables too.
-Please review all abbreviations of DM, DM1, DM2 or other and they are well identified in the manuscript.
- Line 355-464: Please collect all authors’ citations at the Reference section as indicated in the “Instructions for Authors” and adjust at the line.
Castro-Rodríguez M, Carnicero JA, Garcia-Garcia FJ, Walter S, Morley JE, Rodríguez-Artalejo F, et al. 355 Frailty as a major factor in the increased risk of death and disability in older people with diabetes. J Am Med Dir Assoc. 2016;17(10):949–955. 357
Saeedi P, Salpea P, Karuranga S, Petersohn I, Malanda B, Gregg EW, et al. Mortality attributable to 358 diabetes in 20–79 years old adults, 2019 estimates: Results from the International Diabetes Federation 359 Diabetes Atlas, 9th edition. Diabetes Res Clin Pract. 15 de febrero de 2020;108086.
(accessed on ….2020).
-I recommend this manuscript to observe all the specifications that remain to be adapted: “Inspiratory Muscle Training in Intermittent Sports Modalities: A Systematic Review” https://doi.org/10.3390/ijerph17124448

Author Response

ITEMIZED LIST OF THE REVIEWERS COMMENTS

Manuscript ID: ijerph-847417

We would like to thank the Reviewers for their thoughtful and constructive comments. We have considered all suggestions and have incorporated them into the revised manuscript. Changes to the original manuscript are with “track changes”. We believe our manuscript is stronger as a result of the modifications. An itemized point-by-point response to the Reviewers’ comments is presented below. 

REVIEWER'S REPORT

This manuscript talks about the Diabetes disease and as dietary evaluation is of utmost  importance for health promotion, disease prevention and individual treatment plans in patients with diabetes. Method: exhaustive search was carried out in various databases that evaluate the dietary intake of people with diabetes mellitus type 1 and/or 2 according to COSMIN and PRISMA standards. Conclusions: The FFQ is the best instrument available to 25 assess dietary intake in people with diabetes.

The comments are described below:

Reviewer 1 comment´s:

-Line 77: Table 1. Search strategy.

I recommend that authors place the information as if it were a table following the “Instructions for Authors”, or as a figure, but this does not fully understand the information collected when searching. You can consult other manuscripts published in the same journal that serves as a reference.

Authors’s Answer: Thank you very much for the time and effort spent reviewing this document. According to your comment. We have added the Search Strategy as a Table.

-Line 170-182: Where is the [38] reference??

Authors’s Answer: Thank you very much for your suggestion. The [38] reference appears in Line 174. It would be possible that it was a mistake about the file. We have ensured that the cite is correctly included.

-Line 204: Please, organize all tables in a homogeneous format as table 3 or 4.

Authors’s Answer: Thank you very much for your suggestion. According to your comment we have changed the information included in Table 2 in a homogeneous format.

Table 2. Studies characteristics and description of status measurement instrument in DM.

-Line 215: Table 3. Methodological quality of the studies and quality of results reported per measurement property, instrument and study.

Authors’s Answer: Thank you very much for your suggestion. We have made the suggested changes in all Tables following the COSMIN guidelines.

-Line 220: Table 4. Summary of findings per measurement property and instrument.

Authors’s Answer: Thank you very much for your suggestion. We have made the suggested changes in all Tables following the COSMIN guidelines.

-Table 3 and 4: Aguiar et al. (2013), there isn´t in the References Section

Authors’s Answer: Thank you very much for your comment. We have added this study in the References Section

-Please verify that all the authors' citations are in the references section with their corresponding numbering, and in the tables too.

Authors’s Answer: Thank you very much for your comment. We have revised the entire manuscript and we have updated the references.

-Please review all abbreviations of DM, DM1, DM2 or other and they are well identified in the manuscript.

Authors’s Answer: Thank you very much for your suggestion. We have revised the entire manuscript and we have made the suggested modifications.

- Line 355-464: Please collect all authors’ citations at the Reference section as indicated in the “Instructions for Authors” and adjust at the line.

Castro-Rodríguez M, Carnicero JA, Garcia-Garcia FJ, Walter S, Morley JE, Rodríguez-Artalejo F, et al. 355 Frailty as a major factor in the increased risk of death and disability in older people with diabetes. J Am Med Dir Assoc. 2016;17(10):949–955. 357

Saeedi P, Salpea P, Karuranga S, Petersohn I, Malanda B, Gregg EW, et al. Mortality attributable to 358 diabetes in 20–79 years old adults, 2019 estimates: Results from the International Diabetes Federation 359 Diabetes Atlas, 9th edition. Diabetes Res Clin Pract. 15 de febrero de 2020;108086.

(accessed on ….2020).

Authors’s Answer: Thank you very much for your suggestion. We have updated the reference section following your suggestion.

-I recommend this manuscript to observe all the specifications that remain to be adapted: “Inspiratory Muscle Training in Intermittent Sports Modalities: A Systematic Review” https://doi.org/10.3390/ijerph17124448

Authors’s Answer: Thank you very much for your suggestion. We have revised the recommended manuscript and we have make a similar adaptation to this following the specification of COSMIN methodology.

Reviewer 3 Report

Dear authors,

Congratulations for your excellent work, below you can find a few suggestions that can help you to improve the manuscript.

Introduction

There is no health cost of diabetes and its management, this is essential to justify the authors' work. Please, correct it.

Line 42: “…there are a wide 41 variety of validated instruments…” needs references. Please, correct it.

Line 45: “…great variety of instruments…” needs references. Please, correct it.

Line 51-53: Objective do not follow PRISMA standards, PICOS is incomplete; there are no reference to interventions, comparisons, and outcomes. Please, correct it.

Materials and Methods

If the authors are going to use the COSMIN guidelines for quality appraisal, please provide greater description of the components of the checklist. Without this Table 3 becomes quite confusing as it’s not clear why only some tools are included, whilst many have measurements of reliability and validity.

 Results

Figure 1. “records after duplicates removed” must be in the right side like the others boxes with studies excluded.  Please, correct it.

Table 2.  It is very complicated to read and consequently understand it, the texts are too long, it looks like a draft (some abbreviations for example are repeated within the table ...), it is not useful for the reader. This table does not invite you to read it but quite the opposite, please summarize. Summarizing does not mean omitting information. I ask you to use your wits to summarize all the information without omitting. This table needs deep work.

Discussion

In general there is significant duplication of results from the tables, while the authors describe the psychometric properties of individual scales, there is no discussion as to why differences might occur. It would therefore be interesting to elaborate on the discussion of why some scales are demonstrated greater psychometric properties than others. 

There is a lack of comparison with other studies (other systematic reviews with the same aim but based in other diseases or populations, for example), exposing the common aspects and the differences between them. Note that the discussion is the most interesting section of the document (it is also the hardest part to draw, in my opinion).

Author Response

ITEMIZED LIST OF THE REVIEWERS COMMENTS

Manuscript ID: ijerph-847417

We would like to thank the Reviewers for their thoughtful and constructive comments. We have considered all suggestions and have incorporated them into the revised manuscript. Changes to the original manuscript are with “track changes”. We believe our manuscript is stronger as a result of the modifications. An itemized point-by-point response to the Reviewers’ comments is presented below. 

REVIEWER'S REPORT

Congratulations for your excellent work, below you can find a few suggestions that can help you to improve the manuscript.

Introduction

There is no health cost of diabetes and its management, this is essential to justify the authors' work. Please, correct it.

Authors’s Answer: Thank you very much for the time and effort spent reviewing this document. According to your comment. We have added this information as follows: “Diabetes is a challenging public health problem, with serious health consequences and healthcare costs associate to a unhealthy lifestyle (vision loss, amputation, renal failure and cardiovascular disease).”

Line 42: “…there are a wide variety of validated instruments…” needs references. Please, correct it.

Authors’s Answer: Thank you very much for your suggestion. We have added the corrected references.

Line 45: “…great variety of instruments…” needs references. Please, correct it.

Authors’s Answer: Thank you very much for your suggestion. We have added the corrected reference.

Line 51-53: Objective do not follow PRISMA standards, PICOS is incomplete; there are no reference to interventions, comparisons, and outcomes. Please, correct it.

Authors’s Answer: Thank you very much for your comment. To make the present systematic review, we are followed the COSMIN methodology for systematic reviews of Patient‐Reported Outcome Measures (PROMs) (published on February 2018) about objectives and searched including 1. Construct of interest (dietary intake), 2. Population (diabetes) and 3. Type of instruments (questionnaires), so the PICOS methodology was not used. We have specified this information in the manuscript.

Materials and Methods

If the authors are going to use the COSMIN guidelines for quality appraisal, please provide greater description of the components of the checklist. Without this Table 3 becomes quite confusing as it’s not clear why only some tools are included, whilst many have measurements of reliability and validity.

Authors’s Answer: Thank you very much for your comment. We have added this information in the Inclusion/exclusion criteria as follows: “at least one measurement property of those reflected in the COSMIN checklist (content validity, structural validity, internal consistency, reliability, measurement error, hypotheses testing for construct validity, cross-cultural validity/measurement invariance, criterion validity and responsiveness)”.

 Results

Figure 1. “records after duplicates removed” must be in the right side like the others boxes with studies excluded.  Please, correct it.

Authors’s Answer: Thank you very much for your comment. We have make the suggested change in the Figure.

Table 2.  It is very complicated to read and consequently understand it, the texts are too long, it looks like a draft (some abbreviations for example are repeated within the table ...), it is not useful for the reader. This table does not invite you to read it but quite the opposite, please summarize. Summarizing does not mean omitting information. I ask you to use your wits to summarize all the information without omitting. This table needs deep work.

Authors’s Answer: Thank you very much for your suggestion. We have shortened the Table 2, with a homogeneous format and avoiding full sentences.

Discussion

In general there is significant duplication of results from the tables, while the authors describe the psychometric properties of individual scales, there is no discussion as to why differences might occur. It would therefore be interesting to elaborate on the discussion of why some scales are demonstrated greater psychometric properties than others.

Authors’s Answer: Thank you very much for your suggestion. We have added this information as follows: “The differences of values of psychometric properties of the included instruments could be explained by the fact that in each study the setting and the characteristics of the population were different. Although these differences, the different questionnaires included different questions about dietary intake that could influence in the reliability and validity values.”

There is a lack of comparison with other studies (other systematic reviews with the same aim but based in other diseases or populations, for example), exposing the common aspects and the differences between them. Note that the discussion is the most interesting section of the document (it is also the hardest part to draw, in my opinion).

Authors’s Answer: Thank you very much for your suggestion. We have included a new paragraph about this issue include two systematic review that have analysed the nutritional status tools in different populations.

Round 2

Reviewer 3 Report

Line 51-53: Objective do not follow PRISMA standards, PICOS is incomplete; there are no reference to interventions, comparisons, and outcomes. Please, correct it.

Authors’s Answer: Thank you very much for your comment. To make the present systematic review, we are followed the COSMIN methodology for systematic reviews of Patient‐Reported Outcome Measures (PROMs) (published on February 2018) about objectives and searched including 1. Construct of interest (dietary intake), 2. Population (diabetes) and 3. Type of instruments (questionnaires), so the PICOS methodology was not used. We have specified this information in the manuscript.

Reviewer’s answer: in this case, you must change it form the 2.1. Design section (line 60), where you mention PRISMA standards .

Table 2.  It is very complicated to read and consequently understand it, the texts are too long, it looks like a draft (some abbreviations for example are repeated within the table ...), it is not useful for the reader. This table does not invite you to read it but quite the opposite, please summarize. Summarizing does not mean omitting information. I ask you to use your wits to summarize all the information without omitting. This table needs deep work.

Authors’s Answer: Thank you very much for your suggestion. We have shortened the Table 2, with a homogeneous format and avoiding full sentences.

Reviewer’s answer: this table still needs work, I expuse some examples:

add ALL the abreviations and its meaning at the end of the table (footnotes) and leave only the abreviations in the table

you repeat abreviations (Exploratory factor analysis(EFA))

you used frecuently "Significant correlation", you can add a symbol (*) and expalin it in the footnotes 

more summary is possible, one example:"α Cronbach between 0.59-0.90" is the same than "α=0.59-0.90", do it across all the tables please in all cases possible, use more the footnotes...

a paragraph with 10 lines is inadequate for a table (the last one), please sumarize it

...

please follow my suggestions in all the table, not do only the examples I present. I am sure you can do it better. Thank you.

Author Response

We would like to thank the Reviewers for their thoughtful and constructive comments. We have considered all suggestions and have incorporated them into the revised manuscript. Changes to the original manuscript are with “track changes”. We believe our manuscript is stronger as a result of the modifications. An itemized point-by-point response to the Reviewers’ comments is presented below. 

REVIEWER'S REPORT

Reviewer 3:

Line 51-53: Objective do not follow PRISMA standards, PICOS is incomplete; there are no reference to interventions, comparisons, and outcomes. Please, correct it.

Authors’s Answer: Thank you very much for your comment. To make the present systematic review, we are followed the COSMIN methodology for systematic reviews of PatientReported Outcome Measures (PROMs) (published on February 2018) about objectives and searched including 1. Construct of interest (dietary intake), 2. Population (diabetes) and 3. Type of instruments (questionnaires), so the PICOS methodology was not used. We have specified this information in the manuscript.

Reviewer’s answer: in this case, you must change it form the 2.1. Design section (line 60), where you mention PRISMA standards .

Authors’s Answer: Thank you very much for your suggestion. We have made the suggested changes.

Table 2.  It is very complicated to read and consequently understand it, the texts are too long, it looks like a draft (some abbreviations for example are repeated within the table ...), it is not useful for the reader. This table does not invite you to read it but quite the opposite, please summarize. Summarizing does not mean omitting information. I ask you to use your wits to summarize all the information without omitting. This table needs deep work.

Authors’s Answer: Thank you very much for your suggestion. We have shortened the Table 2, with a homogeneous format and avoiding full sentences.

Reviewer’s answer: this table still needs work, I expuse some examples:

add ALL the abreviations and its meaning at the end of the table (footnotes) and leave only the abreviations in the table

you repeat abreviations (Exploratory factor analysis(EFA))

you used frecuently "Significant correlation", you can add a symbol (*) and expalin it in the footnotes

more summary is possible, one example:"α Cronbach between 0.59-0.90" is the same than "α=0.59-0.90", do it across all the tables please in all cases possible, use more the footnotes...

a paragraph with 10 lines is inadequate for a table (the last one), please sumarize it

please follow my suggestions in all the table, not do only the examples I present. I am sure you can do it better. Thank you.

Authors’s Answer: Thank you very much for your suggestion. We have revised the entire Table 2 and we have improved the writing of this Table.
